# Wide-range soft anisotropic thermistor with a direct wireless radio frequency interface

Mahmoud Wagih [1] ✉, Junjie Shi[2,3], Menglong Li [2], Abiodun Komolafe [2], Thomas Whittaker [4], Johannes Schneider [1], Shanmugam Kumar [1], William Whittow[4] & Steve Beeby [2]

Temperature sensors are one of the most fundamental sensors and are found in industrial, environmental, and biomedical applications. The traditional approach of reading the resistive response of Positive Temperature Coefficient thermistors at DC hindered their adoption as wide-range temperature sensors. Here, we present a large-area thermistor, based on a flexible and stretchable short carbon fibre incorporated Polydimethylsiloxane composite, enabled by a radio frequency sensing interface. The radio frequency readout overcomes the decades-old sensing range limit of thermistors. The composite exhibits a resistance sensitivity over 1000 °C$^{-1}$, while maintaining stability against bending (20,000 cycles) and stretching (1000 cycles). Leveraging its large-area processing, the anisotropic composite is used as a substrate for sub-6 GHz radio frequency components, where the thermistor-based microwave resonators achieve a wide temperature sensing range (30 to 205 °C) compared to reported flexible temperature sensors, and high sensitivity (3.2 MHz/°C) compared to radio frequency temperature sensors. Wireless sensing is demonstrated using a microstrip patch antenna based on a thermistor substrate, and a battery-less radio frequency identification tag. This radio frequency-based sensor readout technique could enable functional materials to be directly integrated in wireless sensing applications.

The use of flexible, soft, and conformable large-area sensors enables pervasive monitoring of physiological parameters in epidermal and implantable sensors[1]. Body area networks (BodyNets)[2] are a key application where soft and stretchable smart sensing materials can be applied to detect body parameters[3,4]. Temperature is one of the most fundamental measurands[5], and an integral sensing parameter of most wearable sensor systems. For instance, soft and flexible sensors are used for monitoring skin temperature, either as individual sensors[6,7], discrete components in a wireless system[8], or in a multiplexed array[9,10]. Temperature sensors are also needed alongside other sensors, such as pH[11] or chemical biomarkers[12], to compensate for temperature cross sensitivities.

With the exception of positive temperature coefficient (PTC) thermistors, flexible temperature sensing materials and sensors exhibit relatively low sensitivity, with a Temperature Coefficient of Resistance (TCR) under 100% °C$^{-1}$ [5–7] (see the Methods section for the TCR calculation formula). Therefore, they require complex sampling circuitry to resolve the small changes in resistance. On the other hand, PTC thermistors exhibit a very high sensitivity and TCR, but only for a very narrow range of temperatures, typically below 10 °C[1]. This has limited their use to sensing systems which do not cover a large range of temperatures[1].

Another common feature in the aforementioned temperature sensors[1,6–12] is their readout mechanism. These flexible and stretchable

[1]University of Glasgow, James Watt School of Engineering, Glasgow, UK. [2]University of Southampton, School of Electronics and Computer Science, Southampton, UK. [3]PragmatIC Semiconductor Ltd., Cambridge, UK. [4]Loughborough University, Wolfson School of Mechanical, Electrical, and Manufacturing Engineering, Loughborough, UK. ✉e-mail: Mahmoud.Wagih@glasgow.ac.uk

soft temperature sensors target a conventional sense, digitise and compute, then wirelessly transmit system. In such systems, the analog sensing waveforms are sampled by an analogue to digital converter (ADC) before being processed[13], and subsequently transmitted wirelessly to gateway such as a smartphone. From a readout circuit perspective, the non-linear response of PTC thermistors has led to complex ADCs being developed to interface with them[14,15].

On the other hand, in low data-rate sensors detecting physical parameters such as temperature and humidity, direct wireless transmission of the measurand could enable more pervasive wireless sensing[16–20]. RF sensing tags or antennas are typically used where there is a clear interaction between the antenna and its surroundings, e.g. dielectric changes due to soil moisture variation or ice accumulation[21,22], or vital signs using wearable antennas[23]. Moreover, RF sensor readout has been widely used based on additional functional sensors including 2D materials[24], conductive polymers[25,26], inorganic sensitive materials[27], lumped sensors[28], and even bacterial cultures[29]. For gas sensing, it was shown that reading the response of resistive sensors at kHz frequencies can improve their linearity and sensitivity[30]. However, it is not clear from current sensing RF applications that an RF (MHz to GHz range) readout could outperform reading the sensor's response at or near DC. Furthermore, most materials used in RF sensing systems are retrofitted to highly sensitive resonators, with no report on material design specifically for RF sensing applications. Therefore, while it is widely recognised that next-generation networks, 6G and beyond, will support sensing through wave-matter interaction[31], functional materials are yet to be developed for this purpose.

Here, we report an RF-enabled thermistor, overcoming the decades-old limitation that is the dynamic range of PTC thermistors, which hindered their adoption in sensing applications, as well as enabling a direct wireless readout through a thermistor-integrated antenna, as shown in Fig. 1a. The low-cost organic thermistor based on a carbon polymer matrix composite (PMC) is formulated to achieve a very high temperature coefficient of resistance (TCR) and is fabricated using a scalable moulding process with 3D printed casts, enabling it to act as a temperature-controlled microwave absorber. Instead of manufacturing a miniaturized sensor with read-out electrodes, sampled at low or near-DC frequencies, the large-area PMC thermistor is used as a substrate for resonant microwave components, enabling its response to be remotely read without digital conversion at the sensor.

## Results

### Operation principle: material design for RF sensing

The proposed sensor relies on the temperature-modulated absorption of RF waves propagating through the thermistor composite.

Thus, information about the temperature can be directly modulated onto the amplitude of the radiated power, as illustrated in Fig. 1a. As seen in the simplified equivalent circuit model in Fig. 1b, of a guided-wave transmission line (using the simple LC model) feeding a resonant antenna, the PTC thermistor acts as a temperature-modulated dielectric loss term $R_T$, which reduces the radiated power, represented by the radiation resistance $R_{radiation}$. Moreover, temperature changes affect both the conductivity and real permittivity. Thus, the capacitance, $C_T$, of a transmission line or an antenna will be temperature-modulated, allowing the sensor to be used in a versatile manner as either an amplitude or phase-based sensor, or using the shifts in the resonant frequency.

While the ADC in a traditional DC or near-DC read-out will be able to resolve the PTC thermistor's response to a high resolution over a narrow temperature range, the thermistor's response above a certain temperature falls outside its readout range, as illustrated in Fig. 1c, based on the experimental data detailed in Supplementary Note 1. On the other hand, the RF response of the material will never be fully lossless (i.e. near-zero conductivity), due to the presence of the lossy CFs. As a result, the material will have an observable RF resistivity over a wider temperature range than the DC readout approach.

Given the drastic change in the resistance of PTCs in response to temperature, should a PTC composite be used as a substrate for an antenna, its total radiated power and gain will be highly dependent on temperature. Figure 1d,e shows the power density radiated from a microstrip patch antenna using the proposed thermistor as a substrate. It can be clearly seen that the low substrate resistance at low temperatures inhibits the radiation, where a temperature increase transforms the substrate into behaving as a low-loss dielectric, leading to maximum radiation; Supplementary Note 2 details the full-wave electromagnetic simulation parameters used to obtain the simulated electric field response.

### Thermistor composite preparation and characterisation

The proposed thermistor is based on Polydimethylsiloxane (PDMS) as a low-cost, moldable, soft, stretchable, and flexible PMC. The PDMS acts as a temperature-sensitive binder which swells under high

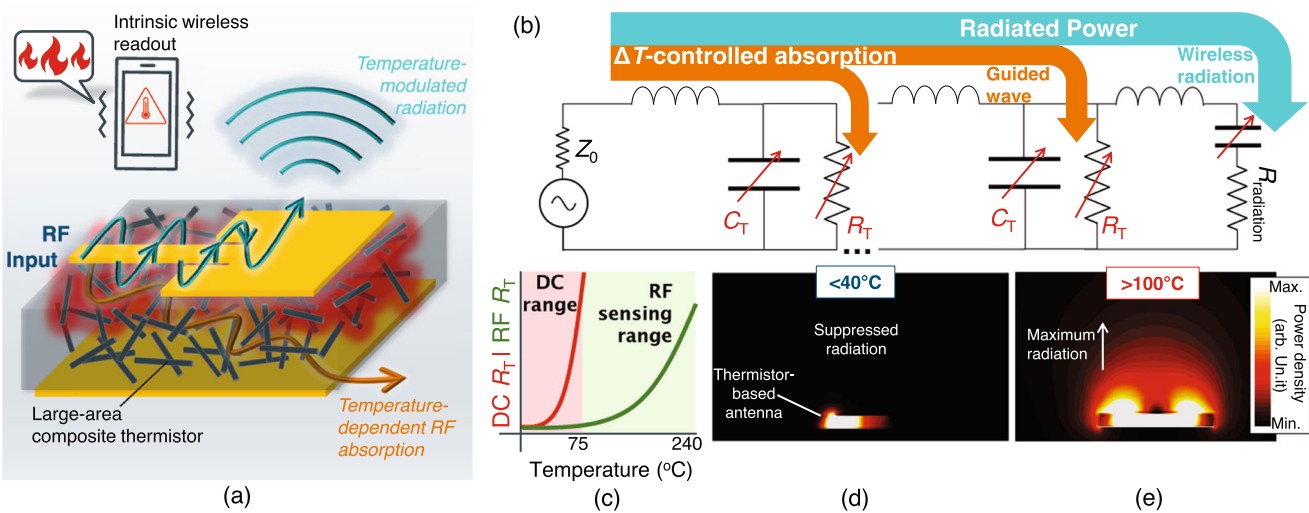

**Fig. 1 | RF-enabled intrinsic wireless temperature sensing. a** Illustration of the PMC PTC's sensing application in a temperature-modulated antenna; **b** circuit model of a transmission line and an antenna using the thermistor as a substrate, showing the temperature-dependent power density radiated from the antenna; **c** an illustration of the thermistor's resistance response and the detection range of the DC and RF readouts (Supplementary Fig. 1 shows the exact measured values); **d, e** simulated RF power flow around a thermistor-based antenna at low and high temperatures, respectively.

temperatures, increasing the separation between the electrically conductive micro-fibers. Carbon-based PTC thermistors were previously demonstrated with high TCR using carbon nanotubes (CNTs)[32,33] and carbon-based conductors[5]. The proposed thermistor is based on off-the-shelf milled carbon fibre (CF). CF was chosen as the conductive filler due to their low-cost and availability, along with the biocompatibility of CF[34]; the cost of milled CF is approximately 1000× lower than CNTs (see Supplementary Note 14). The composite was formulated using the process described in the Methods section and shown in Supplementary Fig. 3.

The Scanning Electron Micrograph (SEM) of the composite can be seen in Fig. 2a, for a cross-section of the cured composite. The electron energy loss spectroscopy (EELS) contrasts the PMC and a pristine PDMS sample, showing a higher carbon concentration, as seen in Fig. 2b. The CF-to-PDMS ratio can be varied at the mixing stage (see Methods) to vary the conductivity, TCR, and stretchability of the composite; detailed SEMs of the different formulations are shown in

Supplementary Figs. 3 and 4. The cross-sections of the sample (Supplementary Fig. 5) show that the CF takes a layered structure, with a higher concentration leading to higher electrical conductivity in-plane, with additional gaps between the fibres out-of-plane, reducing the conductivity in the Z direction.

Samples of 30 × 15 × 3 mm (XYZ) size were used to characterise the DC response of the thermistor; the in-plane resistance was measured in the X direction. The PMC exhibits a PTC thermistor response due to the binder-conductor interaction, as shown in Fig. 2c. The thermistor's sensing range varies as a function of the carbon fibre ratio; higher CF loading requires a higher temperature to increase the separation between the fibres. The response is also consistent for both heating and cooling, and over at least 14 heating and cooling cycles (Supplementary Fig. 10).

The change in temperature sensitivity for different PMC formulation is not monotonic. In the temperature range under consideration, the rise in resistance of the composites, as illustrated in

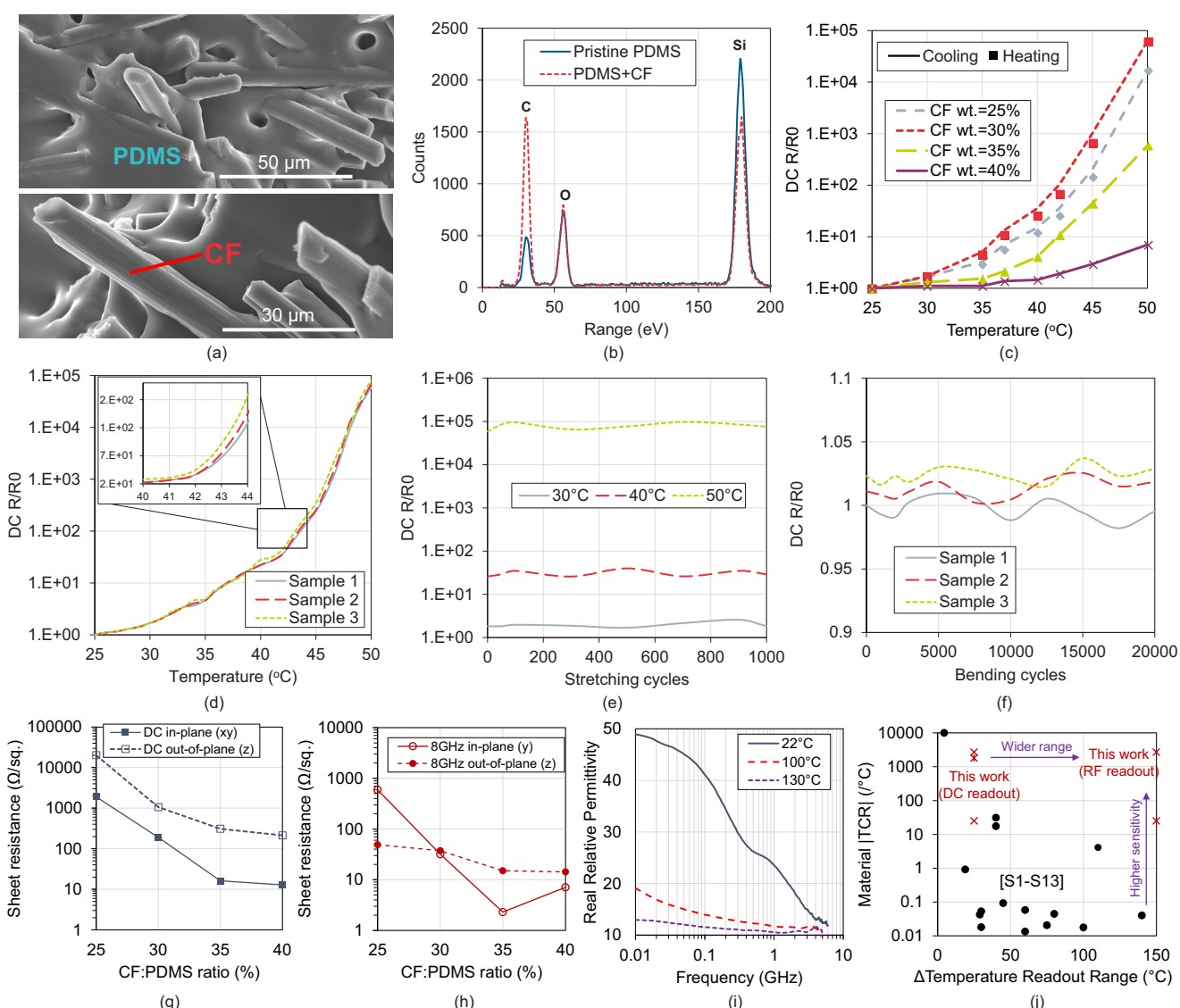

**Fig. 2 | Material composition and characterization. a** SEM micrograph of the CF/PDMS composite showing the distribution of the CFs in the PDMS matrix.; **b** elemental composition of pristine PDMS and the CF/PDMS composite; **c** the DC resistance of the samples of varying 35 wt% CF/PDMS ratios; **d** Repeatability of DC resistance-temperature response of 35 wt% CF/PDMS composite; **e** the resistance stability over 1000 stretching cycles; (**f**) bending resilience over 20,000 cycles; **g** the anisotropic sheet resistance of the composite at DC; **h** the anisotropic sheet resistance of the composite at 8 GHz; **i** the measured real relative permittivity of the material at different temperatures; **j** comparison with previous flexible, printed, or solution-processed thermistors (Supplementary Note 7 details the compared references).

Fig. 2c, is primarily influenced by the conductive CFs embedded within the PDMS matrix. Typically, these CFs exhibit an increase in electrical resistance as the temperature rises. Consequently, the SCF/PDMS composite follows a similar trend up to a specific weight fraction, which, in this case, is 30 wt%. Beyond this weight fraction, the rate at which resistance increases is notably slower compared to the composites with lower CF content. An increased sensitivity at lower temperatures was previously observed in lower CF concentrations[35] which is in line with our results.

The observed exponential change in resistance is reproducible across different samples, of the same composition, as shown in Fig. 2d, and is also repeatable over different cycles and days, as shown in Supplementary Fig. 8. The PMC is resilient to both repeated stretching and bending, benefiting from the hyper-elastic response of the PDMS binder, as seen in Fig. 2e, f, respectively; previous stretchable PTC thermistors were characterized for under 200 stretching cycles[36]. The samples withstand withstand 20,000 bending cycles (over a 6 mm radius) and 1000 stretching cycles (at 70% strain) with no noticeable change in their resistance, indicating their ability to withstand a harsh operation environment; previous soft thermistors were only characterized for up to 200 stretching cycles[36].

For up to a 3.25 cm bending radius, the temperature-resistance response of the thermistor remains unchanged, with under 10% deviation for bending around a radius of 1.75 cm, as shown in Supplementary Note 7. When stretched, the resistance drops at higher temperatures, widening the temperature sensing range, due to the compression in the Z-plane (out-of-plane); see Supplementary Note 7 for the detailed response. Variations in the relative humidity also have minimal influence (under 1%) as shown in Supplementary Note 7. Thermogravimetric analysis (TGA) and Differential Scanning Calorimetry (DSC), detailed in Supplementary Note 8, evidence that the material can be used beyond 300 °C, which covers the full RF sensing temperature range, discussed in the next section. The Coefficient of Thermal Expansion (CTE) of the composite is analytically calculated in Supplementary Note 9.

The room-temperature DC sheet resistance of each composite formulation demonstrate the anisotropic characteristic of the material/the material's anisotropy, as shown in Fig. 2g. The temperature-resistance sensitivity is also anisotropic (as shown in Supplementary Fig. 9) which is attributed to the naturally larger gaps between the fibres in the Z direction, leading to reduced temperature sensitivity out-of-plane.

In addition to the DC response, the GHz conductivity was measured, as detailed in the Methods section, for the in-plane and out-of-plane conductivities, and is shown in Fig. 2h. The waveguide measurement setup and the broadband microwave conductivity are detailed in Supplementary Figs. 11 and 12, respectively. Both the RF and DC results clearly show the material's anisotropy, and that for the higher CF concentrations, the in-plane conductivity is significantly higher than its out-of-plane counterpart, which is due to the alignment of the CF in the casting direction (the XY plane, shown in Supplementary Fig. 5). Nevertheless, in-plane, the material's sensitivity can be observed for a microstrip line aligned in both the X and Y directions (see Supplementary Note 13), meaning that the orientation of any transmission lines, antennas, or resonators, will not significantly influence the sensor's performance.

The different composites exhibit a TCR between 10 and 1000 in the 25–50 °C observed in Fig. 2c, summarised in Supplementary Table 1; the maximum strain that each composite can withstand is also included. Moreover, the 40% CF composite exhibits a TCR exceeding 2000 °C$^{-1}$, when considering its response up to 80 °C, as shown in Supplementary Fig. 9. Therefore, the 40% CF composite is chosen for use in RF sensing, which is also due to having the lowest sheet resistance in-plane, at DC, resulting in higher losses when used an antenna substrate. Figure 2i shows the broadband real relative permittivity $\varepsilon$

(dielectric constant) of the 40% CF composite, measured using a microstrip line; Supplementary Note 11 presents detailed material characterisation results up to 26 GHz, and including temperatures from 22 °C to 240 °C.

Figure 2j compares the TCR of the proposed PMC thermistor with other flexible and printable temperature-sensitive materials. For our DC readout, the temperature sensing range (ΔTemparature) is limited by the 100 MΩ range of the multi-meter (see Methods); using the RF readout, the observable Δtemparature can be improved by over five-fold, using the approach detailed in the next section. Supplementary Note 7 presents a detailed comparison of the sensing range.

## RF temperature sensing using guided waves

Unlike reported highly-sensitive thermistor composites[1], the proposed PMC could be rapidly scaled up to very large areas and moulded to custom geometries. Thus, it can be used as a substrate for RF components at Ultra-High Frequency (UHF) bands or higher. As seen in Fig. 2i, the PMC thermistor demonstrates a high sensitivity over its ΔTemperature sensing range. However, by characterizing a PTC thermistor at DC[5], the ΔTemperature sensing range is significantly limited by the resolution and dynamic range of the sampling ADC. High-value resistive sensor readout interfaces have previously been reported but using high-complexity multi-amplifier circuits[37], which would be incompatible with a large-area soft sensor[4]. Moreover, most resistive sensors exhibit limited accuracy in the MΩ range[38]. Furthermore, using a standalone ADC interface is still bound to the standard sense-compute-communicate flow of conventional sensors[13], without a direct wireless readout.

Harnessing the scalability of the developed large-area PTC thermistor, the composite is moulded into the dimensions of a substrate for a microwave resonator, based on a microstrip transmission line. Two resonators are designed, the first resonator was used to quantify the material's sensory response at microwave frequencies; Supplementary Fig. 13a shows the dimensions of Resonator 1. As depicted in Fig. 3a, the two-port microstrip resonator acts as a band-stop filter at the frequency $f_1$ where the electrical length of the open-ended stub is a quarter of the guided wavelength $\lambda/4$, as well as its higher order harmonics. The magnitude of the in-band rejection ($S_{21}$) is dependent on the losses in the material-under-test[39]. The extracted dielectric properties of the material ($\varepsilon_r$, $\sigma$), resulting in the observable RF sensing response, are summarized in Supplementary Notes 11 and 12, from 10 MHz to 26 GHz.

The resistivity changes in the substrate can be detected through the forward transmission magnitude ($|S_{21}|$) at $f_1$, with the relative permittivity detectable through the phase response or the resonant frequency $f_1$[40]. Various planar and non-planar resonators can be used to observe such response including split-ring resonators[39] and substrate-integrated waveguide[41]. We use the simplest resonator structure, the open-ended quarter-wave line. The resonator was implemented on the 40% CF composite, which is due to it having the lowest resistivity at low temperatures (see Supplementary Table 1), which would lead to the highest dielectric losses at low temperatures.

The two-port vector transmission properties (phase and magnitude) were measured versus temperature for the thermistor-based resonator (Resonator 1, Supplementary Fig. 13a). The experimental setup is described in the Methods section and is shown in Supplementary Fig. 14. Figure 3b shows the broadband response of Resonator 1 at different temperatures. At room temperature and up to 40°, it can be seen that the resonator maintains a very low quality (Q)-factor around its fundamental tone $f_1$. Moreover, the second-order resonance is completely masked by the high losses in the substrate, which rise up to −40 dB at 4 GHz. As the temperature increases, the Q-factor of the first resonance increases and, when the thermistor's resistance increases by over 1000×, based on Fig. 2c, the second order resonance

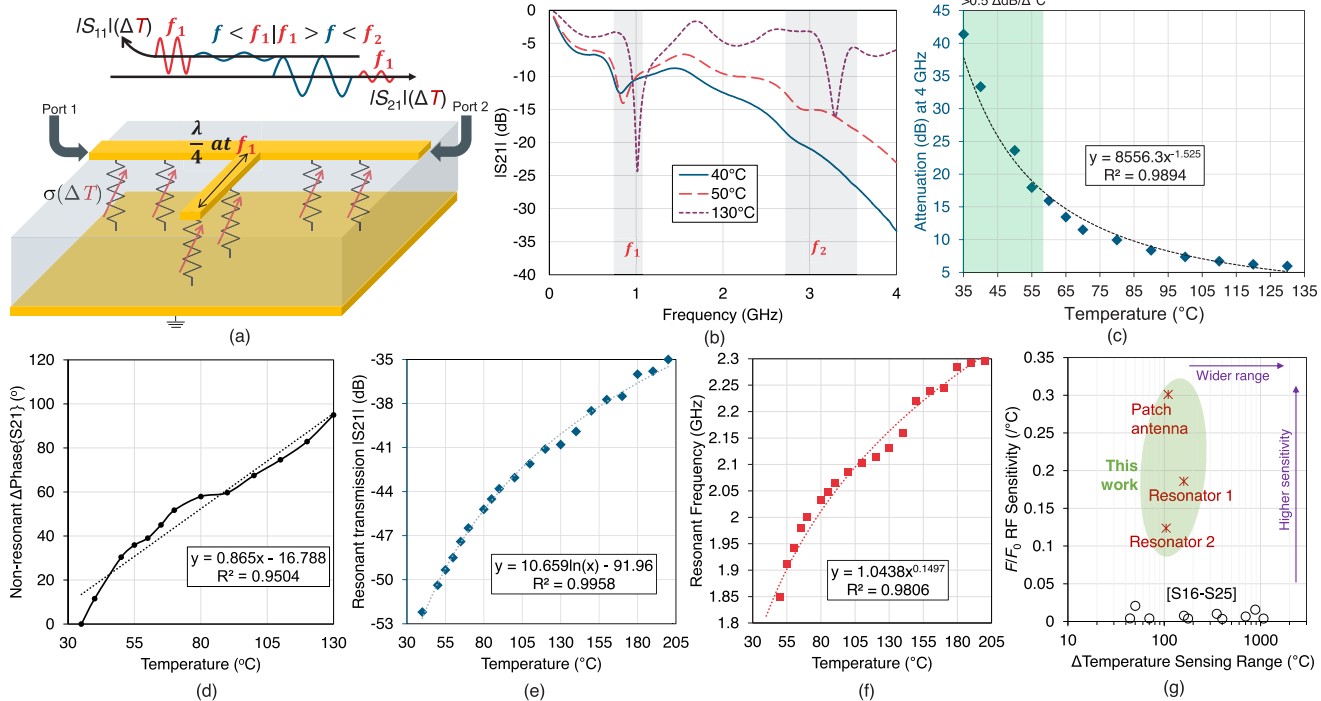

**Fig. 3 | RF-enabled resonant and non-resonant sensor readout using guided waves. a** Illustration of the reflection and transmission mode sensing using an open-ended resonator. **b** Frequency domain response of the thermistor-based resonator for varying temperatures. **c** The non-resonant response of the RF transmission magnitude, over temperature, and the associated conductivity of the substrate at 4 GHz. **d** The non-resonant phase response, at 4 GHz, of resonator 1 over temperature. **e** Wide dynamic range sensing using the resonant magnitude of resonator 2. **f** Wide dynamic range sensing based on the resonant frequency of resonator 2; **g** comparison with previous RF temperature sensors (Supplementary Note 14 details the comparison and the references).

is observed above 3 GHz. The resonance shift and also the magnitude changes up to 130 °C are shown in Supplementary Fig. 13b. The resonator exhibits an average sensitivity of 0.14 dB/ °C and 2.144 MHz/ °C, with a very high $R^2 > 0.99$, as shown in Supplementary Fig. 15.

The non-resonant response of the structure, at 4 GHz, is shown in Fig. 3c. In the temperature range between 35 °C and 55 °C, it can be seen that the $|S_{21}|$ varies by over 20 dB, yielding a high sensitivity exceeding 0.5 dB/ °C. Furthermore, across the tested temperatures, the curve-fitted non-resonant transmission response at 4 GHz exhibited a determination coefficient $R^2 > 0.989$. The phase response of the $S_{21}$, over the same temperature range, is shown in Fig. 3d; the phase delay variation is due to the change in the real permittivity, seen in Supplementary Fig. 20. While the phase of the $S_{21}$ varies with temperature exhibiting a uniform, and almost linear, sensory response, the sensitivity and $R^2$ do not match that of the magnitude. Furthermore, sampling the phase requires a vector receiver which increases the complexity of a sensing system based on the proposed thermistor. Supplementary Fig. 16 shows the broadband phase and magnitude response of the microstrip resonator between 3 and 4 GHz at all measured temperatures.

Observing the higher sensitivity at higher frequencies (see Supplementary Fig. 20 for the extracted permittivities over the sub-6 GHz range), a second resonator (Resonator 2) was designed to resonate around 1.8 GHz before loading; the dimensions of the second resonator are shown in Supplementary Fig. 17. Resonator 2 was characterized in the same setup, but for a wider temperature range, around its first-order resonance. The resonator exhibits an observable sensing range that is over 150 °C wider than the DC readout range of the same composite, as seen in Fig. 3e. By reading the sensing resonator's magnitude at its fundamental resonant frequency, the fitted $S_{21}$ exhibits an $R^2 > 0.99$. An alternative readout mechanism is through the resonant frequency, whose results are shown in Fig. 3f, and are in line

with the observed temperature-permittivity dependence detailed in Supplementary Note 11.

The proposed thermistor-based resonators also exhibit a very high RF sensitivity in their resonant frequency, attributed to the relative permittivity changes seen in Fig. 2i. The sensitivity of the resonator ranges from 2.1 to 3.2 MHz/ °C, as detailed in Supplementary Note 14. Most reported RF temperature sensors were characterised using their resonant frequency[42]; the frequency sensitivity of the proposed resonator is compared to the state-of-the-art in Fig. 3g, where a significant sensitivity improvement can be observed. Supplementary Note 14 provides a detailed numerical comparison including resonant frequencies and resonator topologies, as well as discusses the limits on the temperature sensing range.

## Direct wireless sensing
To realize a wireless sensor, an antenna could be implemented using the thermistor as a substrate or a superstrate. Based on the highly sensitive response of the two-port microstrip resonator to temperature, a microstrip antenna would exhibit high sensitivity to temperature in its radiated power, as shown in Fig. 4a. To explain, a high-conductivity substrate (i.e. the PTC thermistor at low temperatures) would inhibit the radiation due to the field dissipation in the low-resistivity thermistor composite, whereas the high-temperature response would increase the far-field gain due to the reduced dielectric loss in the composite substrate. Thus, temperature could be considered as a method for direct antenna amplitude modulation.

A microstrip patch antenna was designed based on the 40% CF:PDMS composite as a substrate; the antenna's dimensions are shown Supplementary Fig. 18. As with the microstrip resonators, the antenna uses copper films as the conductor to minimize any temperature dependence in the conductive traces and ensure that the observed sensitivity is induced by the dielectric.

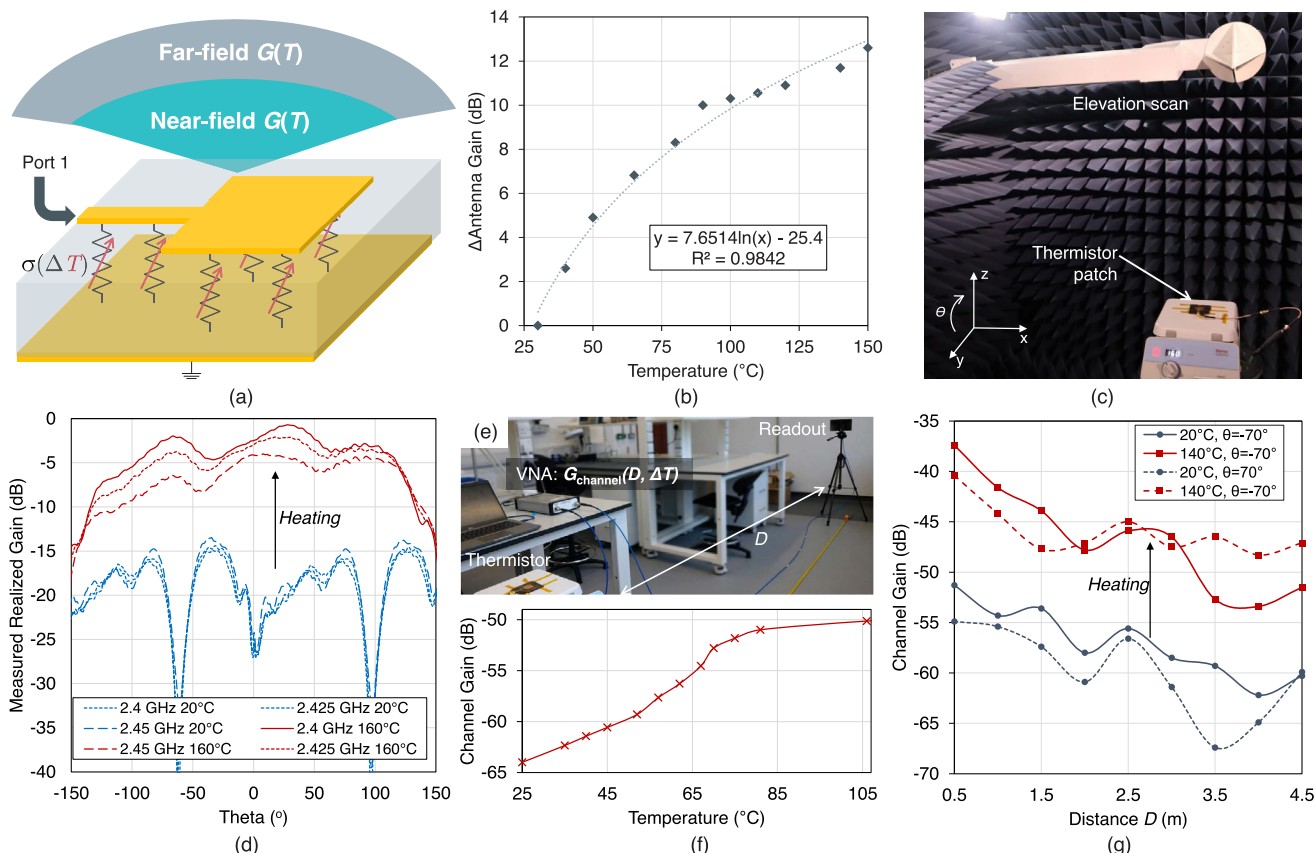

**Fig. 4 | Wireless readout of the thermistor's sensory response. a** The microstrip patch antenna based on the thermistor substrate; **b** measured gain change over temperature, in the antenna's near-field at 2.4 GHz using an *E*-field probe; **c** antenna patterns measurement setup in the anechoic chamber; **d** measured radiation of the antenna in the license-free 2.4 GHz band at 20 °C and 160 °C; **e** the channel gain between the sensing antenna and a reader as a function of temperature; **f** the channel gain as a function of the temperature for a fixed distance between the readout horn and the sensing antenna; **g** the distance-dependent channel gain at different temperatures, in the setup shown in sub-figure **e**.

The antenna's sensory response was observed in the near- and far-field, in both real-world and anechoic environments. The temperature of the antenna was varied and its near-field power was measured in the broadside direction using an electric field probe. The temperature over the surface of the antenna was monitored using an infrared camera as detailed in the Methods section, with the experimental setup shown in Supplementary Fig. 19. Figure 4b shows the change in the antenna's *E*-field, as a function of temperature change. The maximum temperature change corresponds to the temperature at which the connection to the antenna would be damaged by the heat, changing the measured response.

As the near-field magnitude changes over temperature were measured in a single direction, the antenna's temperature-dependent radiation patterns were measured in an anechoic chamber. The antenna's temperature was raised using a hotplate, up to 160°, as detailed in the Methods section; Fig. 4c shows a photograph of the thermistor-based antenna in the anechoic chamber. The antenna's normalized gain patterns are shown in Fig. 4d in the 2.4 GHz license-free band, for both room temperature and the hotplate set to 160°; the measurement source and the patch are co-polarised in the y-direction (i.e. along the patch's microstrip feed). The measured response indicates that the apparent gain change can be measured in all directions across a beamwidth exceeding 120° in the broadside direction, with the sensitivity being dependent on the angle. The patterns were also measured at fixed intervals between 2 and 2.6 GHz, and are shown in Supplementary Fig. 37.

A practical wireless sensor readout would take place in the far-field and over varying angular alignments in a reflective environment, where multiple wireless paths exist. To investigate the combined effect of far-field operation and varying the angular alignment between the wireless readout antenna and the sensor, the sensing patch was evaluated in an indoor echoic environment. The channel gain between the temperature-modulated patch and a standard gain horn was measured, as described in the Methods section. The measurement environment is shown in Fig. 4e and is further detailed in Supplementary Note 13. Figure 4f shows the channel gain as a function of temperature for a 4.9 m separation between the antennas, at 2.501 GHz. The error bars shown represent the observed ±1 dB variation in the channel gain. The measurements are also influenced by fluctuations in the channel gain due to the multi-path propagation; such variations are not seen in near-field interrogations or in an anechoic environment.

The observed far-field response exhibits a sensitivity comparable to that in the near-field. However, the sensor's dynamic range is reduced compared to the near-field operation, in Fig. 4b, and also the resonator's response, in Fig. 3e. This could be attributed to the angular dependence of the gain and also the multi-path effects in the far-field. The broadband response also exhibits a clear temperature-sensitivity as detailed in Supplementary Note 13, and shown in Supplementary Figs. 24 and 25.

Combining the variation of separation and angle variations, the standard gain horn was horizontally moved closer to the thermistor-based patch, varying *D* in the setup from Fig. 4e. Figure 4g shows the

channel gain as a function of distance, where the temperature sensitivity in the channel gain is clearly observed at all distances. As with previous RFID-based sensing systems, the distance dependence can be removed using a reference temperature-insensitive antenna. The ripple due to ground reflections can also be seen in the figure, where it does not affect the sensor's response from being observed, despite operating in a highly reflective environment. The resonant frequency shift of the patch antenna can also be used as a reflection-mode sensor, alongside the wireless gain, and is included in the comparison with the resonators in Fig. 3g. The thermistor patch's $S_{11}$ and resonant frequency, in the same measurement setup, are shown in Supplementary Figs. 26 and 27.

To demonstrate the proposed thermistor composite in a wireless sensing system, a temperature-sensing RFID tag is implemented. A UHF RFID antenna was coated with the thermistor composite and interrogated using a handheld reader, connected via Bluetooth to a smartphone, as illustrated in Fig. 5a; see the Methods section for the RFID tag and reader details. The reader's distance from the tag was limited by the maximum read range at room temperature, i.e. when the thermistor is cold and acting as a shielding surface, significantly reducing the antenna's gain. Only the Received Signal Strength Indicator (RSSI) magnitude is considered to eliminate the need for any phase sampling at the reader. Moreover, as an inexpensive handheld reader was used, the RSSI values are quantised and limited to 1 dB increments.

A reference unloaded RFID tag was located in close proximity to the sensing tag to demonstrate that the RSSI changes are due to the thermistor's response. Figure 5b shows the measured RSSI of both tags as a function of temperature. A highly linear and sensitive region is observed in the 40–65 °C range, with the maximum RSSI change of 10 dB for a 25 °C temperature rise being in line with the observed response for the passive antenna, in the far-field, seen in Fig. 4f, up 70 °C. The reference tag exhibits a ±1 dB RSSI variation which can be

attributed to the non-linearity and temperature dependence of the on-chip RF frontend[43]. The sensor's response is detectable in various channels across the 900–927 MHz band with a comparable response; the measured RSSI changes for different channels are shown in Supplementary Fig. 42.

## Discussion

Our work demonstrates that designing a highly sensitive thermistor composite specifically for RF sensing could demonstrate a significant improvement beyond the state-of-the-art, as well as enable a direct wireless interface with the sensor. This advance is demonstrated with respect to both traditional flexible and solution-processed temperature sensors, read at DC, as shown in Fig. 2i, where the high TCR of a PTC could be read over a significantly wider range compared to traditional DC sensors, through the amplitude of a resonant microwave circuit. As a dielectric sensor, the expansion of the PDMS introduces a very pronounced change in the resonance frequency of both guided-wave resonators and radiating antennas, inducing a sensitivity that is over an order of magnitude higher than prior frequency-domain sensors, operating at the same temperature sensing range, as seen in Fig. 3g. As demonstrated in the RFID-based interrogation, in Fig. 5, the sensor's response can directly modulate the amplitude of a digital wireless signal. This approach can be extended to chipless tags or frequency-selective surfaces (FSS) based on the proposed material as a substrate or a conductor, or integrated within communication antennas for joint communication and sensing in future networks[31].

In our experiments, the sensor was sampled using a bench-top vector network analyzer (VNA), a portable VNA, and a handheld smartphone-connected RFID reader for the thermistor-coated RFID tag. While these circuits exhibit differences in their physical volume and cost, the proposed sensor can be interrogated using fully integrated circuits. On-chip VNAs have been reported with a bandwidth from 10 MHz to over 20 GHz and a comparable dynamic range to commercial bench-top instruments[44]. To enable similar microwave sensors to scale down in dimension, for operation at higher frequencies, new formulations of the material need to be developed and electrically characterised to enable the design of mmWave RF sensing composites. Through mmWave sensing material design and the integration of on-chip readout circuits (i.e. fully-integrated VNAs), a higher spatial resolution could be achieved, leveraging the shorter wavelength of the sensing element.

In summary, we presented an approach to reading the sensory response of large-area soft and stretchable composites. A CF/PDMS composite thermistor was proposed and moulded to large areas enabling it to act as a substrate for RF components including antennas and resonators. It is shown that the RF readout enables the sensor to be read over a wider dynamic range as well as enabling an intrinsic wireless readout, such as using battery-less RFID tags. Moreover, we show that material design for RF sensing outperforms the use of conventional materials in resonant frequency-domain sensors. We believe that future sensitive materials will be scaled up over large surfaces, comparable to the readout wavelength, to enable future large-area sensory electronics to be intrinsically wireless, including chipless sensors as well as direct RF sensor modulation in next-generation wireless networks[31].

## Methods

### CF/PDMS composite fabrication and material characterization

Milled CF powder (FP-MCF-004 Easy Composite, 100 μm length and 7.5 μm diameter) was first mixed with Isopropyl Alcohol (IPA) in a beaker by using Spin mixer (IKA RW16) at spin speed of 200 revolution per minute (RPM) for 10 min. The PDMS is then added to the beaker with the spin speed increased to 300 RPM for 80 minutes to ensure PDMS is mixed. A hotplate is used for heating up the beaker to 140 °C for 80 min to evaporate the IPA; the IPA is added to the CF and PDMS

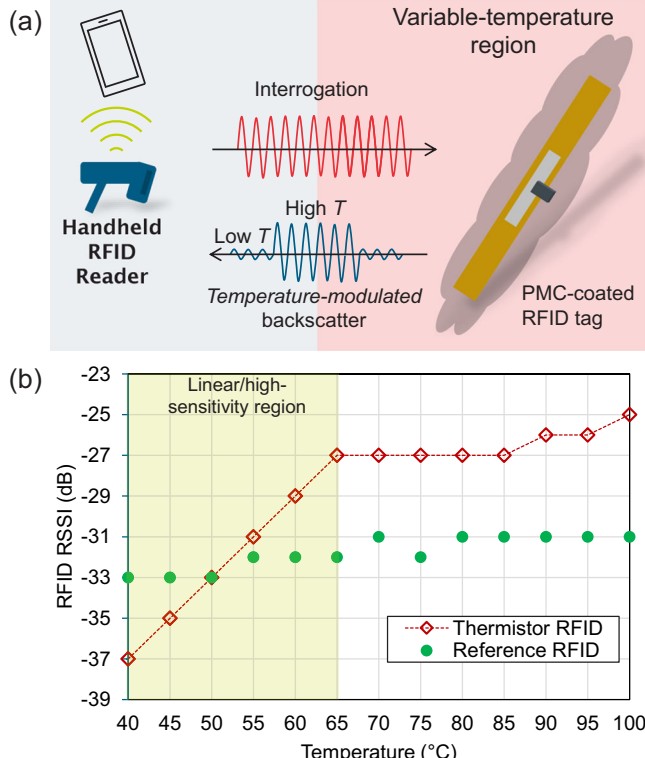

**Fig. 5 | Battery-less wireless readout of a thermistor tag. a** UHF RFID readout using a handheld reader; **b** the temperature-dependent RSSI of the thermistor-coated tag and a reference unloaded tag.

composite for increased fluidity, to ensure CF and PDMS are well mixed. The mixed composite is vacuumed for 30 min at 0 bar to eliminate the air inside the composite, before being cast into the patterned mould and vacuumed for another 60 min to fully eliminate air generated by the molding process; the moulded composite is cured at 80 °C for 90 minutes. The process is summarized in Supplementary Fig. 2. Supplementary Fig. 3 and 4 show detailed SEM micrographs of the PMC for varying magnification and for different CF/PDMS ratios. The SEMs shown in Fig. 2a were taken from a cryogenically-fractured sample.

For both the microstrip resonators and the first microstrip patch antenna, copper tape was used for the conductive traces. The copper was directly attached to the thermistor composites with the RF connectors soldered to the copper traces.

## DC thermistor characterization

Thermistor samples of $30 \times 15 \times 3$ mm dimensions were characterized from 25 to 50°. The samples were placed inside an oven (Thermo Fisher scientific OMS60), with low-resistance leads to a digital multimeter (DMM). The resistance change has been recorded by using Keighley 2001 DMM up to 100 MΩ, and is used to calculate the TCR. The upper temperature limit is imposed by the dynamic range of the DMM. Silver electrodes were added to the sample to ensure repeatable contact with the measurement clips.

The sheet resistance was measured with a 4-point probe system manufactured by Ossilia. The 4 probes are spaced at 1.27 mm apart and are spring-loaded, which avoids piercing the soft samples. The measurement equipment features a micrometre dial which can be used to raise and lower the sample; this was used to ensure the spring-loaded pins were compressed consistently for each sample. The sheet resistance value provided in this work is an average value over multiple measurement cycles; the results from the individual cycles are shown in Supplementary Fig. 5. The sheet resistance value is calculated using the proprietary software provided by Ossilia. The measurement setup using the four-probe system is shown in Supplementary Fig. 6.

## Material thermal characterisation

Differential scanning calorimetry (DSC) measurements were executed utilizing the DSC25 instrument manufactured by TA Instruments. The measurements were conducted across a temperature spectrum spanning from 30 °C to 230 °C, employing a heating rate of 10 °C/min for two heating and cooling cycles, all within a nitrogen atmosphere. Thermogravimetric Analysis (TGA) measurements were performed utilizing the Sdt Q600 instrument, also from TA Instruments. The experiments covered a temperature range extending from 30 °C to 1000 °C, with a heating rate of 10 K/min, and were carried out in an air atmosphere. Density measurements, employing the Archimedes method, were undertaken by immersing the test samples of each composite in water and subsequently quantifying the resultant buoyant force. The density of the samples was subsequently calculated in accordance with Archimedes' principle.

## TCR calculations

The Temperature Coefficient of Resistance (TCR) is calculated for the DC sensing range of the thermistor from room a room temperature of 25 °C. The TCR is calculated as:

$$\text{TCR} = \frac{R_2 - R_1}{R_1(T_2 - T_1)} \quad (1)$$

where $R_1$ is the resistance at $T_1 = 25$ °C and $R_2$ is the maximum resistance reached at $T_2$. The same process was used for all material compositions. The TCR was calculated using the full sensitive range of the thermistor; the same procedure was used to calculate the TCR of the reported thermistors, for the comparison shown in Supplementary Table 2.

## RF conductivity measurement

The RF conductivity was characterised using an X-band waveguide (WR90), measured using an Anritsu MS46522B vector network analyser. Material samples for different CF:PDMS ratios were placed inside a waveguide section and the scattering parameters were measured. The waveguide setup was calibrated with the LRL (line reflect line) calibration method; two different waveguide lengths were used as well as a metal plate for the reflect standard (short). An improved Nicholson-Ross Wier (NRW) method[45] implemented in MATLAB was used extract the material's properties (complex permittivity and permeability) and the conductivity $\sigma$ of the sample was calculated from the electric loss tangent tan$\delta$ using

$$\sigma = \omega \varepsilon' \tan \delta \quad (2)$$

where $\omega$ is the angular frequency and $\varepsilon'$ is the real part of the permittivity of the material.

## Resonator's RF measurements

The s-parameters of the two-port resonator were measured using a Rohde and Schwarz ZVB4 Vector Network Analyzer (VNA), calibrated using a full Through, Open, Short, and Match (TOSM); the VNA sweep was carried out from 10 MHz to 4 GHz with 1501 points. Sub-miniature type-A (SMA) connectors were attached to the copper traces of the resonator for interfacing with the VNA. The resonator was placed on a Thermo Scientific Cimarec hotplate with the temperature varied in 5 °C increments. A Fluke TIS infrared thermal camera was used to monitor the temperature over the surface of the PMC.

The RF sensitivity, in terms of the resonant frequency, of the microstrip resonator was calculated using

$$\text{RF Sensitivity} = \frac{F_1 - F_0}{F_0(T_2 - T_1)^\gamma} \quad (3)$$

where $F_1$ is the maximum resonant frequency (measured at $T_2$) and $F_0$ is the unloaded resonance frequency of the sensor.

## Wireless thermistor characterization

**Resonator sensitivity.** For the gain vs. temperature measurements, the antenna was connected to the port of the VNA, calibrated using the same process as in the two-port resonator measurement. The second port was connected to a near-field probe (R&S HZ series) held at approximately 15 cm in the broadside direction of the microstrip patch. Supplementary Fig. 18 shows a photograph of the measurement setup.

**Radiation patterns.** For the patterns of the microstrip patch antenna (see Supplementary Fig. 18 for the antenna's dimensions), the measurements were carried out in a fully anechoic chamber using a near-field scanner (NSI2000) with a broadband directional log periodic source. The antenna was placed on the positioner with a ThermoFisher hotplate. The measured gain patterns, in Fig. 4d, were normalized to the peak amplitude across all measurements, which is found to be at 2.4 GHz. The normalization to the highest gain value over the different measurements is to capture the gain change in response to temperature, as opposed to evaluating the antenna's own radiation properties.

**Far-field channel gain.** A portable VNA (PicoVNA6) was used to measure the channel gain through the $S_{21}$ between the thermistor patch, on the hotplate, and the readout standard gain horn (Tekbox TBMA4 CISPR16); the horn maintains a gain between 6 and 14 dBi and is matched from 1.5 GHz. The VNA sweep ranged from 100 MHz to 6 GHz, with 2,001 data points; Supplementary Fig. 23 shows the measurement setup and the separation between the antennas. For the distance sweep, the horn was positioned at 50 cm intervals between

50 cm and 450 cm from the heated patch. The height of both antennas was maintained as the distance was swept.

The wireless channel gain measurements are subject to a systematic uncertainty, between 0.5 and 2 dB, depending on the amplitude of the $S_{21}$ measured. This uncertainty is a limitation of the measurement setup (the VNAs) and is detailed in the traceable uncertainty section of the VNA datasheets.

**RFID RSSI-based sensing.** The thermistor-coated RFID tag was based on a compact antenna design[46], previously used for RSSI-based sensing[21]; the reference tag was based on the antenna. The tags were interrogated using a hand-held Zebra RFD8500-EU UHF RFID reader, operating in the 868 MHz license-free RFID band, connected via Bluetooth to a smartphone. Both the sensing tag and the reference loaded tag were placed inside a WKL-100 environmental chamber whose temperature was varied in 5 °C increments. Both antennas were placed on low-permittivity foam spacers of 15 cm height, to separate the tags from the chamber's metallic enclosure. The RFID reader was attached to the glass door of the chamber, approximately 20 cm away from the tags. As the environmental chamber's walls are all metallic, almost emulating a reverberation chamber, it is expected that the RF response will be different from that observed in anechoic and echoic measurements. When interrogating the sensor in the US band (900, 912, and 927 MHz), an Impinj RFID reader was used with an external circularly polarized patch antenna.

### Reporting summary
Further information on research design is available in the Nature Portfolio Reporting Summary linked to this article.

## Data availability
The datasets generated during and/or analysed during the current study are available in the University of Glasgow repository, https://doi.org/10.5525/gla.researchdata.1536.

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

## Acknowledgements
We would like to thank Harry Beeby for his help characterising the RFID thermistor and preparing the datasets and Aran Amin for his help obtaining SEM micrographs. This work was supported by the UK Engineering and Physical Sciences Research Council (EPSRC) under Grant EP/P010164/1 (S.B. and M.W.) and EP/S030301/1 (W.W.), the UK Royal Society under the Research Grant "STEMS" RGS\R1\231028 (M.W), the UK Royal Academy of Engineering and the Office of the Chief Science Adviser for National Security under the UK Intelligence Community Research Fellowship programme (M.W.) and the Royal Academy of Engineering under the Chairs in Emerging Technologies scheme (S.B.).

## Author contributions
M.W. conceptualized the study; J.Sh. and M.L. formulated the materials; M.W., J.Sh., M.L., A.K., T.W., J.Sc., and S.K. conducted the experiments; M.W., J.Sh., M.L., A.K., T.W., J.Sc., S.K., W.W. and S.B. analyzed the results; M.W, W.W., and S.B. acquired funding; M.W. and S.B. supervised the project. All authors reviewed the manuscript. J.Sh. and M.L. contributed equally to the work.

## Competing interests
The authors declare no competing interest.
