## [Peer review file · Nature Communications]

REVIEWER COMMENTS

Reviewer #1 (Remarks to the Author):

In this manuscript, the authors developed a large-area and highly-sensitive thermistor composite specifically for RF sensing, and realized a direct wireless Radio Frequency sensing interface with the sensor. This is a meaningful work that overcomes the decades-old sensing range limit of thermistors. After carefully evaluating, I would like to recommend it to publish in the journal with a major revision. My concerns and comments on this manuscript are followings:

1. As shown in Figure 2(g-h), in most cases, the in-plane conductivity is significantly higher than its out-of-plane counterpart. However, in Figure 2(h), the in-plane conductivity at 25 wt% is lower than its out-of-plane counterpart, and the in-plane conductivity at 30 wt% is approximately same with its out-of-plane counterpart. The authors should specifically discuss the possible reasons for this phenomenon.
2. Why did the authors use two different indicators naming DC resistance and DC R/R0 to characterize the bending and stretching durability of the material? Besides, in Figure 2(f), part of the text "20000" is missing. For the sake of the preciseness of the manuscript, it would be better to exchange the order of mentioning stretching and bending durability to correspond to the order of figures, specifically "The samples withstand 20,000 bending cycles (over a 6 mm radius) and 1,000 stretching cycles (at 70% strain) with no noticeable change".
3. The authors claimed that the thermistor-based microwave resonators achieve the widest temperature sensing range up to 200°C in the abstract. Please supplement the supporting information of exact sensing range.
4. Some grammar errors and typos appear in the current manuscript. In addition, the reference format needs to be carefully revised to fit the journal requirement. Some of the current reference formats are not consistent with each other, such as names, titles and their capitalization.

Reviewer #2 (Remarks to the Author):

The authors present the design and electromagnetic characterization of a temperature sensitive material based on PDMS and embedded carbon fiber. The material is pliable and has improved sensitivity relative to many other studies in the literature. It has good potential in wireless temperature sensing in a broad variety of applications.

Strengths:

1. The paper is well organized, the material fabrication process, the equipment and methodology used to characterize the material has been explained clearly and with good illustrations.
2. Some practical considerations such as how the material holds up to stretching or bending has been examined and presented quantitatively.

Comments worth addressing:

1. One key point that has been alluded to many times in this paper is that the material has a broader temperature range at higher frequencies than at DC. This has also been qualitatively shown in Fig 1(c). I assume this is because there is a limit to the maximum DC resistance that can be measured (100M ohm as listed in this paper) while the RF signal changes can be monitored over a larger temperature range, but this was not very clear to me. Ideally, it would be good if the authors could present Fig 1(c) with actual data so that the concept is clearer.
2. The trends in Fig. 2(c) are non monotonic. CF 30% exhibits the largest overall change with temperature and CF percentages both larger and smaller than this demonstrate smaller changes. Is there an explanation for this?
3. On the subject of Fig. 2(c), CF of 40% demonstrates the least resistance change with temperature so it seems to be the least sensitive of all the CF percentages, at least in the 25-50C range. However,

CF 40% is used for most of the latter half of this paper. From Supplementary note 4, it is explained that this is because of the high TCR of CF40 at 85C, however the corresponding performances of the lower CF samples have not been discussed. Would it be possible to discuss these and why they were eliminated?

4. Given that the conductivity varies in the X, Y and Z direction, would orientation influence the performance of this sensor? For example if an antenna were placed on this material accidentally rotated by 90 degrees.

5. Does relative humidity influence the performance of this material?

6. In Fig 5(b), at what frequency is the RSSI data collected? What is the standard error of the data points, and therefore the statistical significance in the linear region? It would also be good to compare these values to passive UHF RFID temperature sensing solutions such as the Intel WISP, EM 4325 and Farsens systems that offer good temperature resolution using on-board silicon and over a broad range.

I don't think the results are enough for the reader to easily understand (or generalize) the performance in the 1-10 GHz band. I think a more useful way of representing the results would have been to show the change in material properties such as conductivity (as seen in supplementary Fig 12) and dielectric permittivity changes over a larger range of frequencies and for different temperatures. That would give readers maximum flexibility for when it comes to using this material integrated within their own custom antenna designs. For instance, they could use these values as a look up table in simulation to see how an antenna-based sensor would actually perform at different temperatures at a frequency of interest.

Regarding the miniaturization of the antennas to 20 GHz operation: here too, I think the authors would need to show the material properties (as outlined in comment 1 above) for the highest frequency considered for miniaturization (20GHz as listed in the question). The need for this can be highlighted when looking at supplementary figure 12 for example. In Fig. 12, one can see that the conductivity values change quite a bit over the 8-10 GHz range and start trending upwards at 10 GHz. So, one would really need to understand how this material behaves at 20 GHz in order to compute how much signal change to expect when it is integrated with an antenna system.

Reviewer #3 (Remarks to the Author):

Mahmoud Wagih et. al have developed and characterized ultrasensitive sensors for temperature/RF sensing applications. The article is very well written and the images/figures are informative and accurate. In order to be published in this prestigious journal some major revisions need to be taken into account:

1. Please provide thermal expansion coefficient data for nanocomposites. What influence does it play on temperature or RF detection?

2. The physical meaning of the equivalent circuit needs to be further explained.

3. In Figure 2 it is recommended to use dR/R_0 , instead of R.

4. As far as the authors studies, the sensor is highly sensitive to temperature. However, sometimes these sensors can be subjected to mechanical strains, simultaneously with temperature. Can it be used in temperature sensing applications subjected to deformation? Please provide data of the changes of the electrical response with applied strain.

5. Please perform DSC and TGA tests to appreciate the stability of the material up to tested temperatures.

6. The final part of the conclusions should indicate the potential of the materials developed.

In this document, the reviewers comments are copied in black, our **responses are in red**, and the **itemised changes** to the paper are described in **blue**, with pointers to the main and supplementary files. All comments raised by the reviewers have been incorporated in the manuscript.

In response to the reviewers' questions, we have:

- Performed new experiments for evaluating the RF dielectric properties (ϵ_r, σ) of the material, comparing methods and providing broadband temperature-dependent permittivity and conductivity responses (10 MHz to 26 GHz)
- Added new material analysis and characterisation including thermal analysis (TGA/DSC) and analytical calculations of the Coefficient of Thermal Expansion (CTE).
- Carried out further reliability experiments, showing the material's response in variable environments such as changing relative humidities and under tensile strain and deformation.
- Added new repeatability experimental results showing that the RFID sensing response is reproducible and also consistent across frequency channels.
- Improved and expanded the discussion around the material's applications and sensing limitations.

Reviewer #1 (Remarks to the Author):

In this manuscript, the authors developed a large-area and highly-sensitive thermistor composite specifically for RF sensing, and realized a direct wireless Radio Frequency sensing interface with the sensor. *This is a meaningful work that overcomes the decades-old sensing range limit of thermistors. After carefully evaluating, I would like to recommend it to publish in the journal with a major revision.*

We thank the reviewer for carefully reviewing our work, making several constructive suggestions, and for recommending our work for publication following the revisions, all of which have been incorporated in the manuscript.

My concerns and comments on this manuscript are followings:

1. As shown in Figure 2(g-h), in most cases, the in-plane conductivity is significantly higher than its out-of-plane counterpart. However, in Figure 2(h), the in-plane conductivity at 25 wt% is lower than its out-of-plane counterpart, and the in-plane conductivity at 30 wt% is approximately same with its out-of-plane counterpart. The authors should specifically discuss the possible reasons for this phenomenon.

Thanks to the reviewer for raising this question. This is attributed to the difference in the sheet resistance measurement between RF (contact-less), in Fig. 2(h) and DC (contact-dependent), in Fig. 2(g).

At DC, the conductivity of the full sheet can only be observed if the path between the measurement contacts is uninterrupted. As a results, out-of-plane (XZ/YZ direction) the resistance appears higher, which is due to the alignment of the fibres in-plane. Thus, the low-resistance paths in the XY plane are not joined out-of-plane (XZ/YZ direction). This leads to the out-of-plane resistance appearing significantly higher at DC.

On the other hand, the RF conductivity of disconnected layers can be observed due to the “non-contact” interaction of the electric fields with the material’s conductivity. As a result, the presence of the fibres, out-of-plane, meant that the RF absorption is still present (manifesting through the lower sheet resistance).

To demonstrate this, we conducted new experiments by placing two samples back-to-back inside a waveguide, with an insulating low-loss plastic sheet (<0.1 mm-thick) for separation. As seen below, the observed RF conductivity increases with adding thickness to the material.

Changes to the Manuscript:

Supplementary Note 12 has been extended to include the above results (in Supplementary Figure 23) and to discuss the observed conductivity's dependence on the sample's thickness.

2. Why did the authors use two different indicators naming DC resistance and DC R/R0 to characterize the bending and stretching durability of the material?

Absolute resistance was originally used to make evaluating each sample's original resistance easier. However, based on the reviewers' suggestions, this has been now changed to normalised resistance changes, R/R0.

Changes to the Manuscript:

Figure 2 has been revised accordingly, so that the normalised resistance change is shown in stead of the DC resistance (also as later suggested by reviewer 3).

Besides, in Figure 2(f), part of the text "20000" is missing. For the sake of the preciseness of the manuscript, it would be better to exchange the order of mentioning stretching and bending durability to correspond to the order of figures, specifically "The samples withstand 20,000 bending cycles (over a 6 mm radius) and 1,000 stretching cycles (at 70% strain) with no noticeable change".

We thank the reviewer for this suggestion.

Changes to the Manuscript:

The figure presentation has been corrected and the text now appears fully.

The order of the sentence has been corrected accordingly.

3. The authors claimed that the thermistor-based microwave resonators achieve the widest temperature sensing range up to 200°C in the abstract. Please supplement the supporting information of exact sensing range.

We thank the reviewer for this suggestion. The claim in the abstract, which focused on the maximum temperature at which the sensor was characterised is 205°C, as seen in Figure 3(e) and (f). From these figures, the thermistor-based resonator is sensitive to temperature from 30°C to 205°C.

The measured temperature sensing range is further compared and quantified in **Supplementary Tables 2 and 3**, where our RF-enabled thermistor is compared to state-of-the-art temperature sensors, both DC and RF ones.

Changes to the Manuscript:

We have updated the abstract accordingly, to explicitly state the sensing range over which the thermistor was characterised, at RF (35-205°C).

4. Some grammar errors and typos appear in the current manuscript. In addition, the reference format needs to be carefully revised to fit the journal requirement. Some of the current reference formats are not consistent with each other, such as names, titles and their capitalization.

Changes to the Manuscript:

The article has been carefully proof-read and grammar-checking software was used.

The references list has been revised to include the same capitalization. The authors' names formatting has been made consistent, and any further changes required can be made based on the journal's editorial instructions.

Reviewer #2 (Remarks to the Author):

The authors present the design and electromagnetic characterization of a temperature sensitive material based on PDMS and embedded carbon fiber. *The material is pliable and has improved sensitivity relative to many other studies in the literature. It has good potential in wireless temperature sensing in a broad variety of applications.*

Strengths:

1. The paper is well organized, the material fabrication process, the equipment and methodology used to characterize the material has been explained clearly and with good illustrations.
2. Some practical considerations such as how the material holds up to stretching or bending has been examined and presented quantitatively.

We thank the reviewer for carefully reviewing our work and for making several suggestions for major revisions to improve the quality of the manuscript. All comments have been incorporated in the manuscript per the response below.

Comments worth addressing:

1. One key point that has been alluded to many times in this paper is that the material has a broader temperature range at higher frequencies than at DC. This has also been qualitatively shown in Fig 1(c). I assume this is because there is a limit to the maximum DC resistance that can be measured (100M ohm as listed in this paper) while the RF signal changes can be monitored over a larger temperature range, but this was not very clear to me.

Ideally, it would be good if the authors could present Fig 1(c) with actual data so that the concept is clearer.

The sensing range is typically dependent on the readout circuit. DC resistance values can be measured using multi-meters, in a lab setting, or using potential dividers with ADCs in a portable microcontroller-based system. The majority of digital multi-meters however can only resolve DC resistances in the range of 50-100 MOhm, limiting the readout range of the thermistor, and requiring complex ADCs and potential dividers, as found in state-of-the-art approaches.

We have conducted new experiments to extract the material's apparent sheet resistance at RF (2.4 GHz) up to 230°C. These results are now compared in Figure 1(c).

At GHz frequencies, the resistivity of the material changes at a slower rate compared to DC, which allows the temperature changes to be resolved over a significantly wider dynamic range compared to the DC readout. Based on our new experimental data, detailed in Supplementary Note 1, the RF sheet resistance changes are observable over a nearly 5x wider temperature range.

Changes to the manuscript:

We have changed Figure 1(c) accordingly. The figure now shows the measured room temperature and heated resistances (up to 80°C for DC, and 200°C for RF). The data trend has been plotted in Figure 1(c), as the figure is only intended to be a high-level introduction to our motivation; note that the full

numerical comparison of the resistivity over temperature has been added to **Supplementary Figure 1**, in Supplementary Note 1.

2. The trends in Fig. 2(c) are non monotonic. CF 30% exhibits the largest overall change with temperature and CF percentages both larger and smaller than this demonstrate smaller changes. Is there an explanation for this?

In the temperature range under consideration, the rise in resistance of the composites, as illustrated in Figure 2c, is primarily influenced by the conductive short carbon fibres (SCFs) embedded within the PDMS matrix. Typically, these SCFs exhibit an increase in electrical resistance as the temperature rises. Consequently, the SCF/PDMS composite follows a similar trend up to a specific weight fraction, which, in this case, is 30wt.%. Beyond this weight fraction, the rate at which resistance increases is notably slower compared to the composites with lower SCF content.

This can be attributed to the occurrence of percolation in electrical networks at around 30wt.%, resulting in direct contact between the fibres and the formation of an electrical network. As a result, the resistance of the composite decreases. At this stage, the overall change in resistance for the composite is governed by the combined effects of these two competing mechanisms, leading to a reduced rate of resistance increase as the temperature rises at higher SCF loadings.

Changes to the manuscript:

We have updated the main text, in **Lines 155-161**, to explain how increasing the CF percentage increases the sensitivity to temperature, at DC, up to 30wt.%, beyond which the DC sensitivity to temperature drops at lower temperatures.

3. On the subject of Fig. 2(c), CF of 40% demonstrates the least resistance change with temperature so it seems to be the least sensitive of all the CF percentages, at least in the 25-50C range. However, CF 40% is used for most of the latter half of this paper. From Supplementary note 4, it is explained that this is because of the high TCR of CF40 at 85C, however the corresponding performances of the lower CF samples have not been discussed. Would it be possible to discuss these and why they were eliminated?

The 40wt.% CF:PDMS composite was chosen as the RF substrate as it has the highest conductivity and therefore induces the highest loss in the structure. For the lower CF concentrations, the resistance value is too high to induce significant and noticeable RF losses.

This is now clearer in the results shown in Supplementary Note 1, where we compare the change in the DC and the RF (2.4 GHz) sheet resistance (as in the answer to question 1).

Changes to the manuscript:

We have updated Section 2.2, RF Temperature Sensing, in **Lines 221-228**, to explain why the material with the highest conductivity/lowest sheet resistance was chosen as the substrate for the RF components.

New further permittivity characterisation of all the composites has been added to **Supplementary Figure 24**, to show the potential for using the different composites for sensing from 18 to 26 GHz.

4. Given that the conductivity varies in the X, Y and Z direction, would orientation influence the performance of this sensor? For example if an antenna were placed on this material accidentally rotated by 90 degrees.

We thank the reviewer for raising this concern. We have now experimentally demonstrated that this has no effect on the sensor, both from a materials and an RF perspective, as detailed below.

From a **material perspective**, the electrical conductivity characteristics of the SCF/PDMS composite discussed in this study are primarily dictated by the alignment of the fibres within the PDMS matrix. Our manufacturing process resulted in a consistent alignment of the fibres in the in-plane direction, although their orientation within the plane remained random. Consequently, we can anticipate a higher conductivity in the in-plane direction compared to the out-of-plane direction, leading to an anisotropic electrical response. This anisotropic resistance behaviour can directly impact sensor performance. Due to the more random alignment in the XY plane (in-plane), we can expect that the alignment of the sensing structure or resonator, will not alter the performance of the sensor.

From an **RF perspective**, we have conducted new experiments to demonstrate that orientation, in the XY plane, plays a minimal role.

We have verified this by measuring the transmission response through a transmission line aligned in the X and Y directions of a substrate. As seen in the following figure (based on new results), the material maintains comparable permittivity (real) and loss tangent (imaginary permittivity/conductivity) in both directions, as extracted from a microstrip line.

In terms of temperature sensitivity, both microstrip lines were heated to approximately the same temperature (200°C). As shown below, both orientations observe a significant resistance change. The variation in the X and Y direction is primarily due to the alignment of the fibres, leading to a different response, given the high aspect ratio of the fibres.

Further variations could be attributed to the non-homogeneity of the PMC, leading to regions of higher CF agglomeration resulting in an overall higher resistance after heating. Furthermore, temperature variations across both samples during the resistivity measurements are highly likely, as the infrared camera used could only show the temperature at the surface of the material. Further uncertainty could arise from the presence of micron-scale air gaps between the microstrip line and the substrate.

Changes to the manuscript:

We have added the measured RF properties of the substrate in both the X and Y directions to **Supplementary Note 13**. The above graphs have been added as new supplementary figures (**Supp. Fig. 26 and 27**) to show the measured real permittivity and loss tangent in the X and Y direction, using two microstrip lines on the same substrate.

In the paper's main text, we explain that the in/out-of-plane anisotropy does not impose restrictions on the alignment of the antenna/sensor with the fibres, based on the measured response. This is explained in **Lines 209-214**.

5. Does relative humidity influence the performance of this material?

PDMS is indeed humidity-sensitive; our new experimental results demonstrate that humidity plays a minimal role in changing the resistance. To explain, the swelling of the PDMS in response to humidity is insignificant compared to the thermal expansion, caused by the variation in temperature. In order to demonstrate this, we have measured the resistance change in response to humidity. For the 40wt.% CF:PDMS PMC, the new results shown below indicate that a resistance change 0.003 /RH% is observed, at a temperature of 40°C.

To demonstrate that the sensitivity to temperature is maintained across various relative humidity levels, the temperature was swept for very dry conditions (RH<30%) and when the air is saturated with water vapour (RH>80%). As seen below, the temperature sensitivity remains apparent, and almost uninfluenced by the slight resistance change due to the opposing humidity levels tested.

Changes to the manuscript:

The measured resistance change in response to humidity has been added to Supplementary Note 6, and discussed in the main paper in Lines 185-186.

6. In Fig 5(b), at what frequency is the RSSI data collected? What is the standard error of the data points, and therefore the statistical significance in the linear region?

It would also be good to compare these values to passive UHF RFID temperature sensing solutions such as the Intel WISP, EM 4325 and Farsens systems that offer good temperature resolution using on-board silicon and over a broad range.

The RSSI of the RFID tag was measured at 868 MHz. All results were based on 5 takes with the mean data shown in the graph, under +/-1 dB variation was observed in the measured RSSI. The 1 dB uncertainty is due to the limited resolution of the handheld RFID reader, demonstrating how a simple amplitude detector could be used to read the proposed sensor.

To further address the reviewer's question, we interrogated the sensor and the reference at 3 different frequencies (i.e. channels) in the neighbouring RFID frequency band around 900-920 MHz. As seen below, comparable relative changes in the RSSI are observed across all channels, implying that the proposed sensor is not limited to a specific frequency.

We thank the reviewer for highlighting the digital RFID systems capable of sensing temperature.

Our work aims to show that reading the sensor response using RF, rather than reading the response at DC or using a low-frequency capacitive response, then modulating it digitally into a data packet. Our approach is scalable to all resistive sensors and could be integrated as part of linearised systems with a digital readout. All existing RFID-type temperature sensors such as the WISP/Farsens, require the ADC interface. Our approach therefore has the advantage of not adding any further hardware to a simple RF modulator (e.g. a single transistor OOK modulator), allowing wireless sensors to be used in new environments and applications.

Due to this fundamental difference between our analogue/RF sensor, and the indicated digital sensors, is difficult to compare, numerically, the response. For example, in works investigating soft thermistors and temperature-sensitive materials, no direct comparison was presented to digital solid-state sensors, such as [6] and [7]:

- Liu, H., Du, C., Liao, L. et al. Approaching intrinsic dynamics of MXenes hybrid hydrogel for 3D printed multimodal intelligent devices with ultrahigh superelasticity and temperature sensitivity. *Nature Commun* 13, 3420 (2022). <https://doi.org/10.1038/s41467-022-31051-7>
- Hao, S., Fu, Q., Meng, L. et al. A biomimetic laminated strategy enabled strain-interference free and durable flexible thermistor electronics. *Nature Commun* 13, 6472 (2022). <https://doi.org/10.1038/s41467-022-34168-x>

The above works did not directly compare the response of the material with respect to digital temperature sensing ICs, which include the ADC, a linearisation circuit, and a digital/serial communication interface.

The motivation behind our work lies in avoiding such additional circuitry for a wireless sensor, where the amplitude of a radiated signal could be directly modulated using an antenna on the thermistor substrate. While these systems are indeed miniaturized, the additional chip area occupied by the ADC, digital serial communications interface, and associated RF energy harvesting components not only add cost and complexity, but also increase the environmental footprint of the system, which is mostly attributed to ICs [<https://ieeexplore.ieee.org/abstract/document/924514>].

Changes to the manuscript:

We have added the new measured results of the sensor's RFID response over the three different 900-927 MHz frequency channels to **Supplementary Note 21** and discussed in **Lines 466-470**.

Supplementary Note 20 has also been extended to include a note contrasting the proposed sensing approach to the digital RFID temperature sensors highlighted by the reviewer.

I don't think the results are enough for the reader to easily understand (or generalize) the performance in the 1-10 GHz band. I think a more useful way of representing the results would have been to show the change in material properties such as conductivity (as seen in supplementary Fig 12) and dielectric permittivity changes over a larger range of frequencies and for different temperatures. That would give readers maximum flexibility for when it comes to using this material integrated within their own custom antenna designs. For instance, they could use these values as a look up table in simulation to see how an antenna-based sensor would actually perform at different temperatures at a frequency of interest.

We thank the reviewer for this valuable suggestion, which we have addressed through additional experimental results and analysis.

As all proposed structures use the material as a substrate or superstrate for a wireless component (a resonator or an antenna), the apparent permittivity and conductivity was measured using a microstrip line, with the permittivity extracted from the propagation constant of the line. These measurements were performed at different temperatures within our sensing range. Thus, the reader can then design an antenna or a resonator based on the heated response of the material (low-loss under 6 GHz), which can then lead to the antenna's gain increasing when heated for the sensor's response.

The transmission parameters were measured at room temperature and at 180°C. In addition, per our response to the anisotropy and misalignment question, this was repeated in both the X and Y direction, to demonstrate how the sensor's response would change if a resonator or antenna are placed in orthogonal directions in-plane.

The new extracted permittivity is shown below:

The measured data demonstrates that the material's real permittivity is most sensitive to temperature under 1 GHz, while the resistivity (imaginary permittivity) is sensitivity over most of the sub-6 GHz spectrum.

Changes to the Manuscript

We have updated Figure 1(b), showing how the permittivity changes also influence the capacitance, and hence the phase or resonant frequency, of RF components based on our material.

We updated Figure 2, adding Figure 2(i) which shows the real relative permittivity of the material at different temperatures, over MHz to GHz frequencies.

For further clarity and to enable further design based on our material, extensive new experimental data has been added to:

- Supplementary Note 11: showing the measured real permittivity and sheet resistance of the material over different temperatures at:
 - 500 MHz to 19.5 GHz, showing the limit of the temperature sensing bandwidth. (**Supplementary Figures 17 and 18**).
 - 10 MHz to 6 GHz, showing the detailed response of the material over its temperature sensing bandwidth. (**Supplementary Figure 22**).
- Supplementary Note 12: showing the heated and room temperature response:
 - Beyond 20 GHz, where the material exhibits limited sensitivity (**Supplementary Figure 22**).
 - Between 18 and 19.5 GHz, where the microstrip and waveguide frequency bands overlap, allowing the two measurement techniques to be cross-validated (**Supplementary Figure 23**).

Regarding the miniaturization of the antennas to 20 GHz operation: here too, I think the authors would need to show the material properties (as outlined in comment 1 above) for the highest frequency considered for miniaturization (20GHz as listed in the question). The need for this can be highlighted when looking at supplementary figure 12 for example. In Fig. 12, one can see that the conductivity values change quite a bit over the 8-10 GHz range and start trending upwards at 10 GHz. So, one would really need to understand how this material behaves at 20 GHz in order to compute how much signal change to expect when it is integrated with an antenna system.

We thank the reviewer for this related suggestion, which has also guided our additional experiments in response to the previous suggestion.

Per our previous answer, we observed through the broadband heated microstrip line measurements that the material's highest sensitivity is below 6 GHz. Furthermore, we also observe that while there is significant frequency dependence in the dielectric properties, the sensitivity is observable over the full 10 MHz to 6 GHz spectrum.

For completion and to provide the reader with the most complete characterisation, the material's properties were measured when heated and after cooling in the 20-26.5 GHz band. Due to the difficulty in heating the waveguide setup in-situ, the samples were pre-heated to 100°C inside a waveguide section, before being inserted into the measurement jig in under 30 seconds. However, this does not impact our study's outcomes, as the >10 GHz falls outside our temperature sensing operation bandwidth.

The lower sensitivity and overall higher loss beyond 20 GHz could be attributed to the CF filling increasing the conductivity, which leads to a broadband high absorption beyond 6 GHz. Therefore, formulating the composite for mmWave sensing using micro-scale components can only be achieved following a reformulation of the composite. As the scope of our work is to show how the RF readout could overcome the limitations of the traditional DC thermistor read-range, we believe that further optimisation of the composite for mmWave sensing is part of the future work, and is also dependent on the target temperature sensing range and the mechanical properties of the material.

Changes to the Manuscript:

We have added the new measured results up to 19.5 GHz using a microstrip line to **Supplementary Note 11**, alongside the new measured waveguide results from 19 to 26 GHz in **Supplementary Note 12**.

The discussion section has been updated, in **Lines 502-510**, to explain that further material design for different (primarily higher) frequency bands is required and is part of the future work, as it is beyond the scope of this work.

The abstract has been updated to specify that the developed RF sensing components are all sub-6 GHz, and all conclusions now reflect that the material's temperature sensing range is restricted to sub-6 GHz applications.

Reviewer #3 (Remarks to the Author):

Mahmoud Wagih et. al have developed and characterized ultrasensitive sensors for temperature/RF sensing applications. The article is very well written and the images/figures are informative and accurate. In order to be published in this prestigious journal some major revisions need to be taken into account:

1. Please provide thermal expansion coefficient data for nanocomposites. What influence does it play on temperature or RF detection?

We thank the reviewer for their suggestion. We have addressed this question through new experimental and analytical results, to extract the material's coefficient of thermal expansion (CTE).

We have analytically calculated the CTE of the nanocomposite, based on the experimentally measured density of the material (to extract the volume fraction of the PDMS and CF). As CF has a negative CTE, when heated, the conductive fibres will reduce in length which increases the electrical resistance and lowers the dielectric permittivity. Thus, leading to lower losses, dielectric permittivity, and in turn higher resonance frequencies (as observed throughout our RF results, in Figures 3, 4 and 5). As for PDMS, the positive CTE accounts for the exponential increase in resistance in response to temperature, which is harnessed in our work to realise extremely sensitive wireless RF thermistors.

The calculated CTE, shown below, was obtained using conservative estimates for the CF alignment angle, in-plane, and also the anisotropy of the composite.

We wish to highlight that the main contribution of our work is the demonstration of the significantly enhanced readout at RF, instead of DC; we expect these conclusions to be transferable to other thermistor composites, whose CTE could be tailored to enhance the resistivity and permittivity changes for a specific temperature range.

Changes to the Manuscript:

We have added **Supplementary Note 9**, where the CTE data is presented, alongside the details of its calculation, and how it affects the sensor's response. This has been referred to in the main text, in **Lines 191-192**.

The calculated approximate CTE of the composite, for all the CF:PDMS formulations has been added to **Supplementary Figure 14**.

2. The physical meaning of the equivalent circuit needs to be further explained.

The equivalent circuit is used to show the temperature-modulated losses in the composite, and how they influence the radiation of the antenna. The lossy composite, with a temperature-dependent resistance, affects the transmission loss through the substrate, and also the resistance loss when the composite is used as a substrate for an antenna. Therefore, in the schematic of Figure 1(b), two temperature-dependent resistors are included to show how the developed composite could be used in guided-wave sensors and also in radiative antenna-based sensors.

As for the capacitance, it is dependent on the substrate's permittivity and is therefore another sensing mechanism of the sensor. In a non-resonant device, e.g. the transmission line in Figure 3(c) and (d), the permittivity is responsible for the phase changes. Moreover, the change in capacitance is also responsible for the resonance frequency shift observed in Figure 3(f), which outperforms all previous sensors as highlighted in Figure 3(g).

Changes to the Manuscript:

We have re-annotated Figure 2(b) to explain:

- The variable resistance causing the loss in a guided-wave sensor.
- The variable resistance causing the loss in a radiative antenna-based wireless sensor.
- The variable capacitance of the substrate, which causes a resonance shift and phase delay changes, in both guided-wave and radiative antenna-based sensors.

We have revised the main text in **Lines 82-91**, to explain each variable component in the equivalent circuit, referring back to the nanocomposite's properties, and highlighting the physical phenomena responsible for the observed RF electrical response.

3. In Figure 2 it is recommended to use dR/R_0 , instead of R .

We thank the reviewer for this suggestion, which has been implemented accordingly.

Changes to the Manuscript:

dR/R_0 is now used instead of R in Figure 2 in the main text of the manuscript.

4. As far as the authors studies, the sensor is highly sensitive to temperature. However, sometimes these sensors can be subjected to mechanical strains, simultaneously with temperature. Can it be used in temperature sensing applications subjected to deformation? Please provide data of the changes of the electrical response with applied strain.

We thank the reviewer for this suggestion.

We have performed new experiments to explore the effects of strain and bending (causing both tensile and compressive strain), on the electrical temperature-resistance relation. In addition, we have explored the anisotropic resistance changes in-plane and out-of-plane with applied strain, to further explain the observed response.

When the material is subjected to strain, the room temperature resistance rises, in-plane, and drops, out-of-plane. This is attributed to the compression of the composite in the Z direction. The measured response is shown below:

This effect causes a shift in the thermistor's response, where the highest change in resistance is observed at a higher temperature, when strain is applied, as seen below:

We note that most previously-reported flexible thermistors, including stretchable ones, e.g. [37] in the main text, have only been characterized between stretching cycles, as opposed to with applied strain.

On the other hand, bending has a less significant influence on the thermistor's temperature-resistance relation, where no noticeable change is observed for a bending radius of 3.25 cm. For a sharper bending radius of 1.75 cm, a 5°C shift in the saturation temperature of the PTC thermistor is observed, as seen below.

For most conformable wearable or industrial applications, this bending radius could be sufficient. However, for higher bending radii, or for direct applied strain, an additional compensation mechanism will be required to ensure the drift in the thermistor's response is accounted for.

Changes to the Manuscript:

We have added **Supplementary Note 7** to show the new experimental results detailing the resistance change in response to temperature, under both bending and applied strain.

Supplementary Note 7 also includes the material's resistance in both planes for different strain values, showing the anisotropic response.

We have updated **Lines 174-177** to explain the strain limits over which the temperature sensor's response is almost unchanged.

5. Please perform DSC and TGA tests to appreciate the stability of the material up to tested temperatures.

We thank the reviewer for this suggestion. We have performed new DSC and TGA tests, observing the material's response over the full temperature sensing range.

As seen in the TGA results below, the material can be used at temperatures higher than the explored sensing range in our work. Furthermore, over 98% of the sample's weight is maintained over our full sensing range.

Changes to the Manuscript:

We have added **Supplementary Note 8**, to present and discuss the results of the TGA and DSC tests.

We added **Lines 186-188** to discuss the TGA results explaining that the material can be reliably used over its full sensing range.

6. The final part of the conclusions should indicate the potential of the materials developed.

We thank the reviewer for their suggestion. We believe that this study and the developed material will enable future developments in both wireless RF sensors and in RF sensing composites and materials. We believe that future work is required to enable new sensing composites to be used at even higher frequencies, for more sensitive, compact, and highly integrated systems, compatible with the “6G” and beyond frequencies.

Changes to the Manuscript:

We have added **Lines 488-495 and 504-512** to highlight the applications and limitations of our material, and the need for further material development and discovery for RF sensing.

REVIEWERS' COMMENTS

Reviewer #1 (Remarks to the Author):

The authors have addressed the concerns well and I suggest to accept this paper.

Reviewer #2 (Remarks to the Author):

The authors have done a good job with the response document and have highlighted the changes made clearly. Most of my questions/ concerns have been satisfactorily addressed.

I do, however, have two follow up counter-comments for them to consider:

1. The explanation about Fig. 1(c) and the addition of material in the supplementary note is helpful. In addition, I would encourage the authors to at least add the 80 and 200C values in Fig. 1(c) to convey some sense of scale to the reader.
2. About the non-monotonic trend in Fig. 2(c) - I think the explanation provided in the red text in the response document is more descriptive than what has been added to the manuscript. I would recommend the authors use this instead.

Reviewer #3 (Remarks to the Author):

The authors have carried out all the proposed revisions satisfactorily. Publish it as is.

Reviewer #1 (Remarks to the Author):

The authors have addressed the concerns well and I suggest to accept this paper.

We thank the reviewer for their constructive comments and helpful feedback.

Reviewer #2 (Remarks to the Author):

The authors have done a good job with the response document and have highlighted the changes made clearly. Most of my questions/ concerns have been satisfactorily addressed.

We thank the reviewer for their constructive comments and helpful feedback. The further comments have been incorporated in the manuscript as recommended.

I do, however, have two follow up counter-comments for them to consider:

1. The explanation about Fig. 1(c) and the addition of material in the supplementary note is helpful. In addition, I would encourage the authors to at least add the 80 and 200C values in Fig. 1(c) to convey some sense of scale to the reader.

Fig. 1(c) has been updated with the x-axis showing the maximum temperature at which the sheet resistance of the material was measured.

2. About the non-monotonic trend in Fig. 2(c) - I think the explanation provided in the red text in the response document is more descriptive than what has been added to the manuscript. I would recommend the authors use this instead.

As suggested by the reviewer, the discussion of the non-monotonic thermistor response of the composite has been updated in the manuscript, to match the discussion in the response-to-reviewers document.

The discussion in the manuscript now includes:

“The change in temperature sensitivity for different PMC formulation is not monotonic. In the temperature range under consideration, the rise in resistance of the composites, as illustrated in Figure 2(c), is primarily influenced by the conductive CFs embedded within the PDMS matrix. Typically, these CFs exhibit an increase in electrical resistance as the temperature rises. Consequently, the SCF/PDMS composite follows a similar trend up to a specific weight fraction, which, in this case, is 30wt.%. Beyond this weight fraction, the rate at which resistance increases is notably slower compared to the composites with lower CF content.”

With further detailed included in the supplementary file, and accessible through the open peer-review file, we believe that the readers will be able to access a clear and comprehensive explanation of, which does not disturb the flow of the manuscript.

Reviewer #3 (Remarks to the Author):

The authors have carried out all the proposed revisions satisfactorily. Publish it as is.

We thank the reviewer for their constructive comments and helpful feedback.